# A Multicenter Clinical Diagnostic Accuracy Study of SureStatus, an Affordable, WHO Emergency Use-Listed, Rapid, Point-Of-Care Antigen-Detecting Diagnostic Test for SARS-CoV-2

Lisa J. Krüger,[a] Andreas K. Lindner,[b] Mary Gaeddert,[a] Frank Tobian,[a] Julian Klein,[a] Salome Steinke,[a] Federica Lainati,[a] Paul Schnitzler,[c] Olga Nikolai,[b] Frank P. Mockenhaupt,[b] Joachim Seybold,[d] Victor M. Corman,[e,f] Terry C. Jones,[e,f,g] Nira R. Pollock,[h] Britta Knorr,[i] Andreas Welker,[i] Stephan Weber,[j] Nandini Sethurarnan,[k] Jayanthi Swaminathan,[k] Hilda Solomon,[k] Ajay Padmanaban,[k] Ma Thirunarayan,[k] Prabakaran L,[l] Margaretha de Vos,[m] Stefano Ongarello,[m] Jilian A. Sacks,[m] Camille Escadafal,[m] Claudia M. Denkinger,[a,n] for the Study Team

aDivision of Infectious Disease and Tropical Medicine, Heidelberg University Hospital, Heidelberg, Germany
bCharité—Universitätsmedizin Berlin, Institute of Tropical Medicine and International Health, Berlin, Germany
cVirology, Heidelberg University Hospital, Heidelberg, Germany
dCharité—Universitätsmedizin Berlin, Medical Directorate, Berlin, Germany
eCharité—Universitätsmedizin Berlin, Institute of Virology, Berlin, Germany
fGerman Center for Infection Research (DZIF), Charité Partner Site, Berlin, Germany
gCenter for Pathogen Evolution, Department of Zoology, University of Cambridge, Cambridge, United Kingdom
hDepartment of Laboratory Medicine, Boston Children's Hospital, Boston, Massachusetts, USA
iDepartment of Public Health Rhein Neckar Region, Heidelberg, Germany
jAcomed Statistik, Leipzig, Germany
kApollo Hospitals, Chennai, India
lFoundation of Innovative New Diagnostics (FIND), New Delhi, India
mFoundation of Innovative New Diagnostics (FIND), Campus Biotech, Geneva, Switzerland
nGerman Center for Infection Research (DZIF), Heidelberg University Hospital Partner Site, Heidelberg, Germany

**ABSTRACT** Access to reverse transcription-PCR (RT-PCR) testing, the gold standard for severe acute respiratory syndrome coronavirus 2 (SARS-CoV-2) detection, is limited throughout the world, due to restricted resources, available infrastructure, and high costs. Antigen-detecting rapid diagnostic tests (Ag-RDTs) overcome some of these barriers, but independent clinical validations in settings of intended use are scarce. To inform the World Health Organization's (WHO) emergency use listing (EUL) procedure and ensure affordable, high-quality Ag-RDTs, we assessed the performance and ease of use of the SureStatus for SARS-CoV-2. For this prospective, multicenter diagnostic accuracy study, we recruited unvaccinated participants with presumed SARS-CoV-2 infection in India and Germany from December 2020 to March 2021, when the Alpha (B.1.1.7) variant was predominantly circulating. Paired swabs were performed for (i) routine clinical RT-PCR testing (sampling was either nasopharyngeal [NP] or combined NP and oropharyngeal [NP/OP]) and (ii) Ag-RDT (sampling was NP). Performance of the Ag-RDT was compared to RT-PCR overall and by predefined subgroups, e.g., cycle threshold ($C_T$) value, symptoms, and days from symptom onset. To understand the usability, a system usability scale (SUS) questionnaire and ease-of-use (EoU) assessment were performed. A total of 1,119 participants were included in the analysis, of whom 205 (18.3%) were RT-PCR positive. SureStatus detected 169 out of 205 RT-PCR-positive participants, reporting a sensitivity of 82.4% (95% confidence interval [CI]: 76.6% to 87.1%) and a specificity of 98.5% (95% CI: 97.4% to 99.1%). In the first 7 days post-symptom onset, the sensitivity was 90.7% (95% CI: 83.5% to 94.9%), when $C_T$ values were low and viral loads were high. The test was characterized as easy to use (SUS, 85/100) and considered suitable for point-of-care settings, although quality concerns were raised due to visibly contaminated

Address correspondence to Claudia M. Denkinger, claudia.denkinger@uni-heidelberg.de.

The authors declare no conflict of interest.

[This article was published on 6 September 2022 with an error in the abstract. The abstract was corrected in the current version, posted on 27 September 2022.]

packaging of swabs included in the test kits. The SureStatus diagnostic test can be considered a reliable test during the first week of SARS-CoV-2 infection, with high sensitivity in combination with excellent usability.

**IMPORTANCE** Our manufacturer-independent, prospective diagnostic accuracy study assessed clinical performance in participants presumed to have a SARS-CoV-2 infection at three study sites in two countries. We assessed the accuracy overall and in predefined subgroups ($C_T$ values and symptom duration). SureStatus performed with high sensitivity. Its sensitivity was particularly high in the first 3 days after symptom onset and when $C_T$ values were low (i.e., the viral load was high). The system usability and ease-of-use assessment complements the accuracy assessment of the test and highlights critical factors to facilitate the widespread use of SureStatus in point-of-care settings. The high sensitivity demonstrated by the evaluated Ag-RDT within the first days of symptoms, when most transmission occurs, supports the role of Ag-RDTs for public health-relevant screening. Evidence from this study was used to inform the World Health Organization Emergency Use Listing procedure.

**KEYWORDS** SARS-CoV-2, COVID-19, antigen-detecting rapid diagnostic tests, sensitivity, specificity

Antigen-detecting rapid diagnostic tests (Ag-RDTs) are widely available and are used to complement the current gold standard, reverse transcription-PCR (RT-PCR), for diagnosis of severe acute respiratory syndrome coronavirus 2 (SARS-CoV-2). Ag-RDTs usage aims at increasing testing capacities and early isolation of infected individuals to minimize viral spread (1). Especially in the global south, usage of Ag-RDTs has increased, while access to the gold standard, RT-PCR, is limited due to the required infrastructure and personnel and the high cost. With this expanded usage of Ag-RDTs worldwide, the need for highly sensitive yet low-cost tests has increased (2, 3).

In 2020, the World Health Organization (WHO) initiated a global partnership to ensure affordable, high-quality Ag-RDTs for low- and middle-income countries (LMICs) to increase testing and reduce the spread of the virus (4). Currently, four Ag-RDTs are listed for emergency use (EUL) by the WHO: PanBio nasal and nasopharyngeal (Abbott), Standard Q (SD Biosensor), and SureStatus (PMC Private Limited) all meet the minimum standards of >80% sensitivity and >98% specificity (5). However, greater capacities for both the production of high-quality tests and lower prices are necessary to meet the rising demand in the global south (6).

The primary objective of this multicenter prospective accuracy study was to evaluate SureStatus for clinical diagnostic accuracy and ease of use (EoU) and its suitability as a diagnostic tool for global testing strategies. This study represents the first multicenter, manufacturer-independent diagnostic accuracy study for the Ag-RDT SureStatus. The study informed a large-scale investment from the WHO, the Foundation of Innovative New Diagnostics (FIND), and Unitaid, which enhanced production capacities and technology transfer, facilitating the production and delivery of over 250 million low-cost SureStatus tests to LMICs (7). In 2021, more than 10 million SureStatus tests were made available to the global south for under $2.55 per test (7).

## RESULTS

**Clinical diagnostic accuracy.** Patients were enrolled from 26 February to 25 March 2021 in Germany (26 February to 25 March 2021 in Heidelberg; 1 to 24 March 2021 in Berlin) and from 3 December 2020 to 23 April 2021 in India. During the enrollment period, a total of 1,196 eligible participants meting the inclusion criteria were screened for this study. Of these, 1,133 agreed to undergo a second swab for study purposes (Fig. 1). Across all sites, 13 participants were early exclusions due to a lack of patient information available. After data cleaning and the exclusion of one invalid PCR test result, a total of 1,119 participants were included in the analysis.

The clinical and demographic characteristics of the enrolled participants are summarized

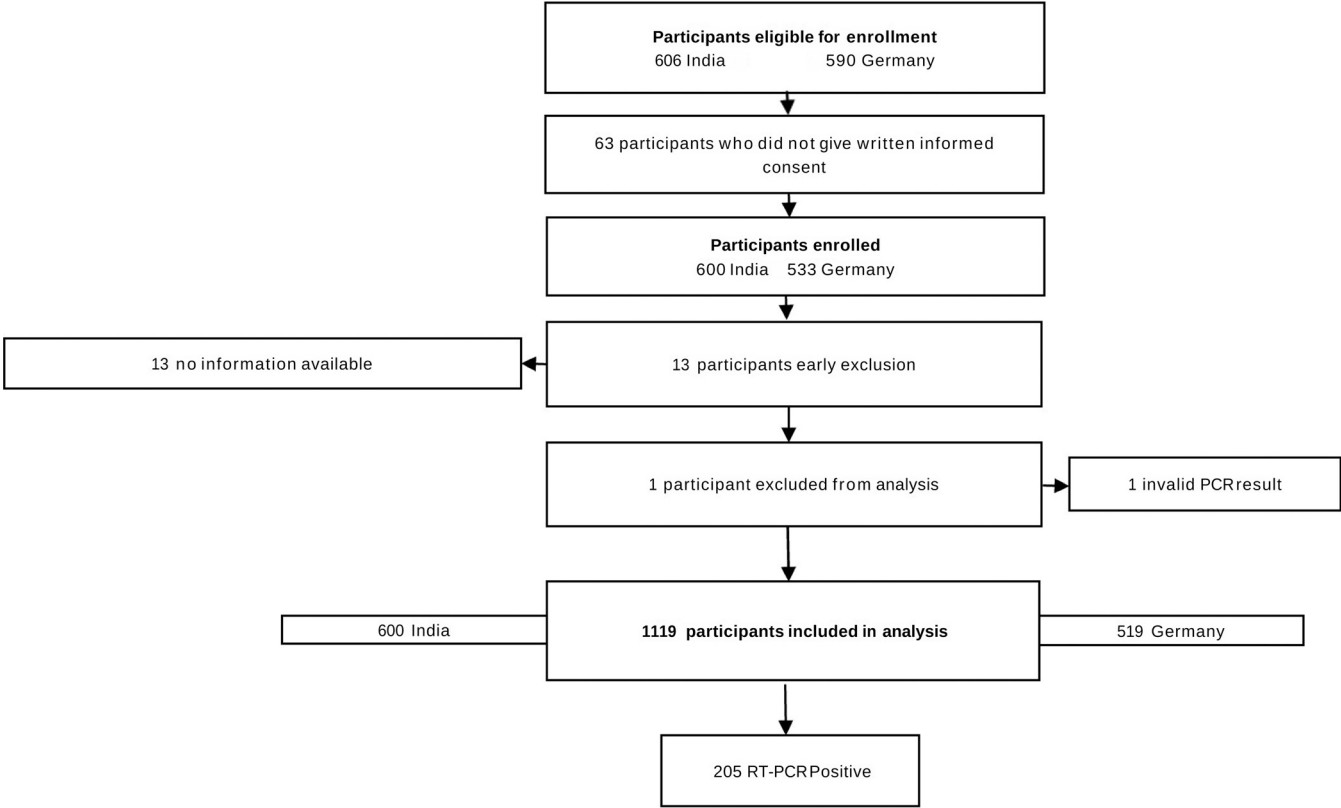

**FIG 1** Study flow.

in Table 1. The median age was similar across study sites (in India, 39 years; interquartile range [IQR], 28 to 52; in Germany, 36 years; IQR, 27 to 50). Overall, 38.3% of participants were female, with a higher female-to-male ratio in Germany (1.4:1) than in India (0.6:1). Of all participants, 20.0% reported comorbidities. In total, 454 participants (40.8%) reported having symptoms on the testing day. However, there were differences between sites; in India, 20.8% of participants presented with symptoms, and in Germany, 64.0% of participants reported symptoms. The median duration of symptoms was 5 (IQR, 2 to 7) days in India and 2 (IQR, 1 to 4) days in Germany. In total, 205 (18.3%) participants were diagnosed with a SARS-CoV-2 infection by RT-PCR testing during the enrollment period, with 19.3% out of 519 participants enrolled in Germany and 17.5% out of 600 enrolled participants in India testing positive. The median cycle threshold ($C_T$) value was slightly higher in Germany, 20.3 (IQR, 17.5 to 23.7), than that in India, 19.0 (IQR, 16.5 to 25.0) (Table 1; Fig. 2).

The SureStatus test had an overall sensitivity of 82.4% (169/205 RT-PCR-positive cases detected; 95% confidence interval [CI]: 76.6% to 87.1%) and a specificity of 98.5% (14 false positives in 914 RT-PCR-negative cases; 95% CI: 97.4% to 99.1%) (Fig. 3). The sensitivity in Germany was 91.0% (91/100 RT-positive cases detected; 95% CI: 83.8% to 95.2%) and in India, 74.3% (78/105 RT-positive cases detected; 95% CI: 65.2% to 81.7%). The specificity in India was 99.6% (2 false positives; 95% CI: 98.5% to 99.9%), compared to 97.1% (12 false positives; 95% CI: 95.1% to 98.4%) in Germany.

Analysis of the performance of SureStatus using $C_T$ values (≤25, ≤30) showed the highest sensitivity in participants with a $C_T$ value of ≤25, 87.7% (95% CI: 78.7% to 93.2%) in India and 97.5% (95% CI: 91.3% to 99.3%) in Germany. In Germany, the sensitivity for participants with a $C_T$ value of ≤30 was 97.8% (95% CI: 92.3% to 99.4%) and for participants with a $C_T$ value of >30, 22.2% (95% CI: 6.3% to 54.7%). In India, the sensitivity was 78.6% (95% CI: 69.5% to 85.5%) in participants with a $C_T$ value of ≤30 and 0.0% (95% CI: 0.0% to 43.4%) in participants with a $C_T$ value of >30. In summary, in Germany, most false negatives had a $C_T$

**TABLE 1** Study population characteristics

| Characteristic | Data for study site(s) | | |
| --- | --- | --- | --- |
| | All | Chennai, India | Heidelberg and Berlin, Germany |
| No. of participants | 1,119 | 600 | 519 |
| Median age (IQR)$^a$ | 37 (27–51) | 39 (28–52) | 36 (27–50) |
| | | | |
| Gender (n [%])$^b$ | | | |
| Female | 427 (38.3) | 176 (29.4) | 251 (48.7) |
| Male | 687 (61.7) | 423 (70.6) | 264 (51.3) |
| | | | |
| Comorbidities (n [%])$^c$ | | | |
| Yes | 224 (20.0) | 99 (16.5) | 125 (24.1) |
| No | 895 (80.0) | 501 (83.5) | 394 (75.9) |
| | | | |
| PCR result (n [%])$^c$ | | | |
| Positive | 205 (18.3) | 105 (17.5) | 100 (19.3) |
| Negative | 914 (81.7) | 495 (82.5) | 419 (80.7) |
| | | | |
| Reporting symptoms (n [%])$^b$ | | | |
| Yes | 454 (40.8) | 125 (20.8) | 329 (64.0) |
| No | 660 (59.0) | 475 (79.2) | 185 (36.0) |
| | | | |
| Median symptom duration (days [IQR])$^d$ | −2 (1–5) | 5 (2–7) | 2 (1–4) |
| Median $C_T$ value (IQR)$^e$ | 20.0 (17.0 to 24.2) | 19.0 (16.5–25.0) | 20.3 (17.5–23.7) |

$^a$IQR, interquartile range.
$^b$Information available for $n = 1,114$ (of 1,119).
$^c$Information available for $n = 1,119$ (of 1,119).
$^d$Information available for $n = 394$ (of 454).
$^e$Information available for $n = 203$ (of 205). Information on two RT-PCR results was not available.

value above 30 with few exceptions, while in India, SureStatus performed less well, with a larger number of false negatives also observed with lower $C_T$ values.

When the test performance was assessed by the duration of symptoms, SureStatus performed well overall in the first 7 days after symptom onset (sensitivity, 90.7%; 95% CI: 83.5% to 94.9%), with declining sensitivity thereafter (>7 days of symptoms: sensitivity, 66.7%; 95% CI: 46.1% to 82.4%). Less than 3 days post-symptom onset, the overall sensitivity was 95.5% (95% CI: 86.8% to 98.5%), with the sensitivity in Germany being slightly higher than that in India, 96.4% (95% CI: 87.9% to 99.0%) versus 90.0% (95% CI: 59.6% to 98.2%), respectively. During the first 7 days post-symptom onset, we found that the SureStatus sensitivity was substantially higher in Germany (sensitivity, 96.2%; 95% CI: 89.3% to 98.7%) compared to that in India (sensitivity, 75.9%; 95% CI: 57.9% to 87.8%). The opposite was shown when SureStatus testing was performed more than 7 days post-symptom onset, although the sample size was smaller and the confidence intervals overlapped (sensitivity in India, 76.9%; 95% CI: 49.7% to 91.8%; sensitivity in Germany, 54.4%; 95% CI: 28.0% to 78.7%).

Out of the 205 total RT-PCR-positive cases, 27 were asymptomatic high-risk contacts. Within this group, the sensitivity of the Ag-RDT was 74.1% (20/27; 95% CI: 55.3% to 86.8%), which was lower than that in the symptomatic participants, 84.2% (149/177; 95% CI: 78.1% to 88.8%). The mean $C_T$ value in the asymptomatic participants was 22.0 (IQR, 16.5 to 26.0), versus 20.7 (IQR, 17.1 to 24.0) in the symptomatic participants.

The high interrater reliability, measured by a kappa result of 1.0, indicates that the test results are clearly interpretable, and no discrepancies were experienced between readers.

**Ease-of-use assessment.** The results of the ease-of-use (EoU) assessment and the system usability scale (SUS) questionnaire are summarized in Fig. 4. SureStatus was scored at 85 out of 100 points on the SUS, indicating a test that is easy to use. Laboratory staff indicated difficulty with applying the required 12 drops to the proprietary test tube. In addition, the handling of the buffer solution, meaning squeezing to apply an appropriate amount (three drops) onto the cassette-formatted test device,

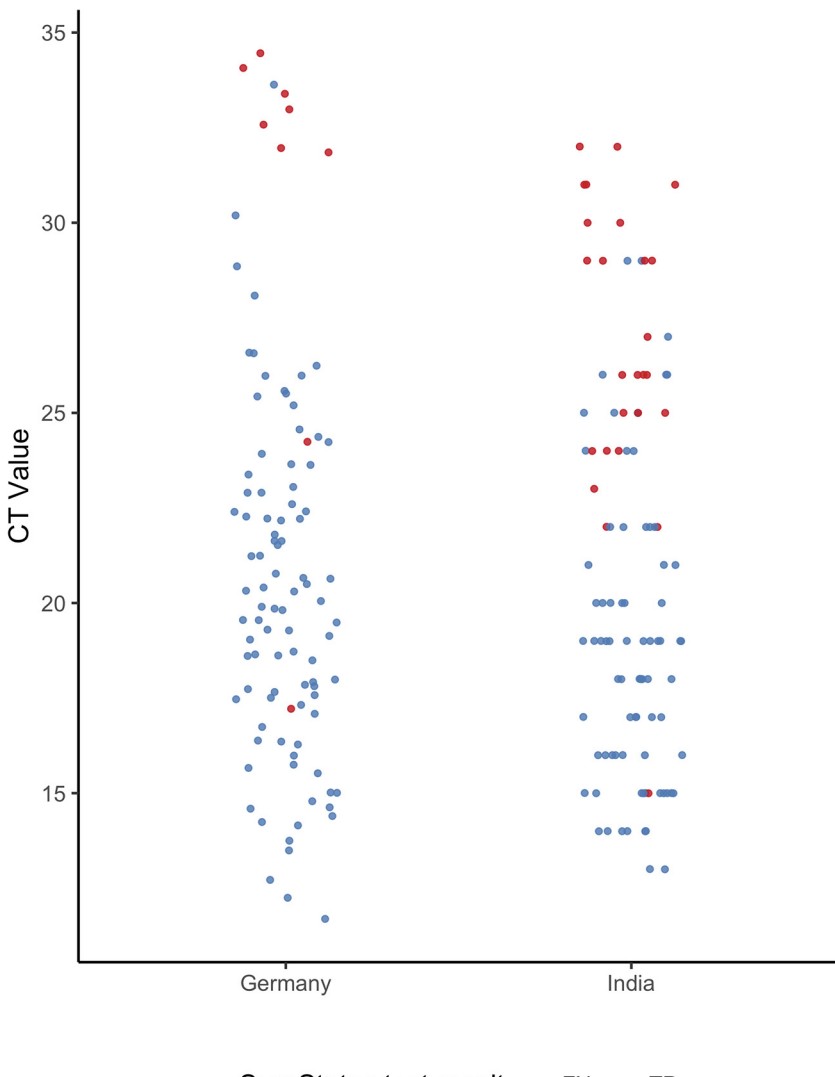

**FIG 2** $C_T$ value comparison between Germany and India. Assays performed in Germany: Allplex SARS-CoV-2 assay from Seegene, Roche Cobas SARS CoV-2 assay on the Cobas 6800 or 8800 systems, SARS CoV-2 assay from TIB Molbiol. Assay performed in India: Thermo Fisher TaqPath COVID-19 combo kit PCR on the Applied Biosystems platform. TP, true positive; FN, false negative.

was considered tedious (Fig. 4). Further, issues were encountered with the quality of the proprietary swabs included in the test device box. Several swabs, only at the German testing sites, were not sterile, as contamination could be seen with the naked eye, and the swabs had to be discarded (on average, five swabs per testing kit). This problem was not experienced at the Indian testing site. In addition, the swabs were often significantly bent in shape and therefore unsuitable for sample collection. These swabs were discarded and not used.

## DISCUSSION

This prospective multicenter clinical diagnostic accuracy study in representative high- and limited-resource settings shows that the SureStatus is a well-performing test with an overall sensitivity of 82.4% and a specificity of 98.5% compared to the reference standard, RT-PCR. The test is easy-to-use and feasible in point-of-care settings. This, combined with the low production costs and high production capacities, justifies the large-scale effort to make the test accessible.

Comparing the accuracy to other lateral-flow assays for SARS-CoV-2, the results were on

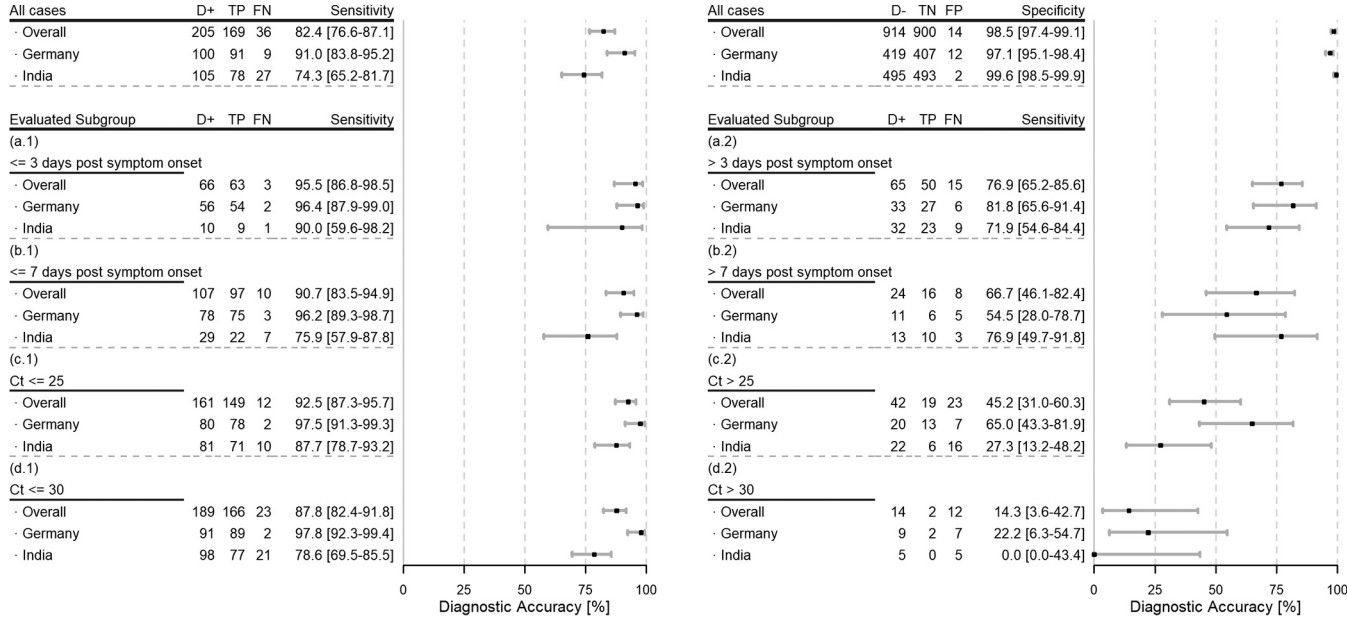

**FIG 3** Overall performance and subgroup analysis post-symptom onset for SureStatus. Overall performance for sensitivity (top left) and specificity (top right); subgroup analysis by duration of symptoms (a.1, ≤3 days; a.2, >3 days; b.1, ≤7 days; b.2, >7 days) and by $C_T$ value (c.1, ≤25; c.2, >25; d.1, ≤30; d.2, >30). TP, true positive; FN, false negative.

par with those of the best performing tests (3). However, there were substantial differences observed in the performance between the two countries. One possible explanation for the lower performance could be differences in the population. In India, a higher number of participants presented without symptoms and in the second week of disease. Furthermore, the low performance in the group with <7 days of symptoms suggests underreporting or possibly different perceptions of the duration of symptoms in different cultures. Also, the impact of environmental conditions (i.e., high temperature and high humidity) could be considered (8, 9). Of note, FIND repeatedly observed lower test sensitivity in the evaluations performed at the study site in India. A root-cause analysis exercise, which included sampling and testing parameters, was conducted and did not return a conclusive cause for the lower performance compared to that at the other study sites. The higher temperature and humidity conditions observed in India are also considered to have contributed to the mixed performance (10). Furthermore, we acknowledge that different test systems were used for PCR testing, and variability between the test systems might have contributed to the differences between sites. Calibration against standards was not possible (11). Nevertheless, with a performance of 90.7% within the first 7 days post-symptom onset, the majority of transmission-relevant SARS-CoV2 infections are likely to be detected by the test, supporting recent published literature (3). The lower performance in the asymptomatic participants is in line with other reports (3) and likely due to the fact that more asymptomatic participants are captured in the second half of their illness, as viral kinetics and thus $C_T$ values in the first week of infection between asymptomatic and symptomatic adults are expected to be the same if tested at the same time (12).

Given that the evaluation was performed when mostly the Alpha variant was circulating (13), the question remains to what extent the findings can be translated to novel variants. While comparable results have been shown for Delta, data are also now emerging for Omicron that suggest that Ag-RDTs targeting nucleocapsid protein remain sensitive (14).

Considering the test's ease of use and rapid turnaround time, along with its high specificity, it could be considered for several use cases on a global scale at low cost. (i) Mass screening, (ii) entry-testing to protect (e.g., in high-risk settings such as hospitals), (iii) testing to release (e.g., contact testing), and (iv) testing to enable (e.g., regular school or workplace

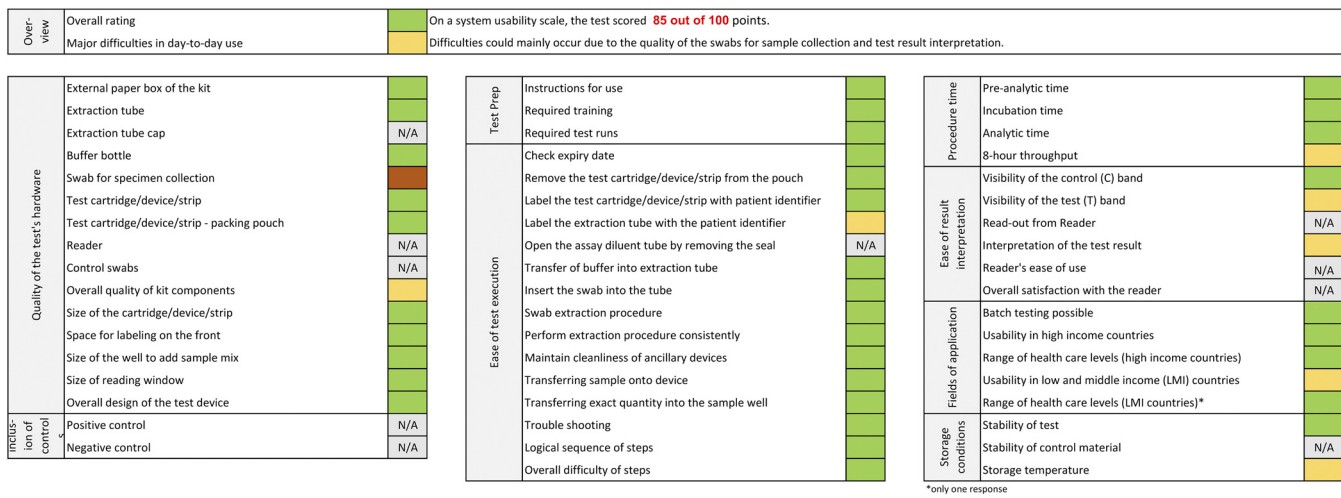

**FIG 4** System usability scale questionnaire and ease-of-use assessment results.

testing) have been suggested in different studies, in addition to (v) the use in symptomatic patients when RT-PCR is not available, such as in low-resource settings, or in combination with RT-PCR, when a rapid decision is necessary. Furthermore, given that self-sampling from the anterior nose (AN) is a reliable alternative to professional nasopharyngeal sampling, scale-up of testing could be possible without requiring large numbers of trained health care workers, provided the result obtained with combined nasopharyngeal and oropharyngeal (NP/OP) sample collection here can be confirmed with AN sampling (15, 16).

Overall, our study has several strengths. Primarily, the population was enrolled in two representative settings in India and Germany, with a population enrolled that showed a broad spectrum of clinical disease (from asymptomatic with high-risk contacts to severely ill). SureStatus was performed at point of care (POC), thus mimicking the real-world challenges of POC testing in three different settings with different levels of resources. The ease-of-use assessment highlighted important points for operationalization of the test and suitability of use in low-resource settings.

However, the study also had several limitations. First, the prevalence of positive cases, and the percentage of symptomatic participants varied substantially between the countries. This likely reflects different phases of the pandemic but also could relate to differences in patient behavior and recommendations for testing. Furthermore, the differences in the reference standard, PCR, introduced limitations to the interpretability of the analysis by $C_T$ value. We also acknowledge that the RT-PCR reference standard has its limitations, as it is not always a meaningful test when considering viable virus and risk of transmission. Lastly, our study excluded vaccinated individuals and those with prior infections, given the uncertainty around the possibility of breakthrough infections at the time of enrollment. Lower viral loads in the case of breakthrough infections, especially in the second week, are likely to diminish the sensitivity of SureStatus in a vaccinated population overall; however, the tests should continue to detect the individuals with the highest viral loads, who are most likely to transmit the virus (12, 17).

In summary, the favorable ease-of-use results, low production costs, and limited infrastructure required for the Ag-RDT testing procedure, in addition to the high sensitivity for infections in the first week of illness, can empower control of population transmission on a global scale, if implemented in well-designed testing and screening programs (18–20).

## MATERIALS AND METHODS

**Ethics statement.** This study was approved by the ethics committees of Heidelberg University Hospital (registration number S-180/2020) and Charité University Hospital, Berlin (EA1/371/20) in Germany and by the

Institutional Ethics Committee for Bio Medical Research of Apollo Hospitals, Chennai, India (IEC application number AMH-C-S-032/09-20). The study was registered in the German Clinical Trial Registry (DRKS00021).

**Clinical diagnostic accuracy.** This study is reported following the Standards for Reporting Diagnostic Accuracy (STARD) (21).

**(i) Study design and participants.** This manufacturer-independent study was conducted in partnership with FIND, the WHO collaborating center for coronavirus disease 2019 (COVID-19) diagnostics. The study was conducted in India at a tertiary-care hospital in Chennai (Apollo Hospitals) and in Germany at two sites: (i) in Heidelberg at a SARS-CoV-2 drive-in testing center managed by the local health department (Rhein-Neckar Region) and (ii) in Berlin at an ambulatory SARS-CoV-2 testing facility of the Charité University Hospital. Participants were eligible for enrollment if aged ≥18 years, determined to be at risk for a SARS-CoV-2 infection by the local health department based on proven contact with a confirmed SARS-CoV-2 case, or having symptoms suggestive of infection, in accordance with WHO criteria (see Table S2 in the supplemental material). Exclusion criteria for the study were as follows: prior positive RT-PCR test for SARS-CoV-2 at any time during the pandemic, vaccination against SARS-CoV-2, inability to provide written informed consent due to limited knowledge of language (in Germany, German or English; in India, English or Tamil), hemodynamic instability, inability to provide a respiratory sample, and recent history of excessive nose bleeds.

**(ii) Study procedures.** Individuals presenting for routine testing and meeting the inclusion criteria were invited to participate in the study. After providing written informed consent, participants first underwent a routine swab for RT-PCR, directly followed by the study-specific swab for Ag-RDT testing, performed by trained study teams. Sampling for RT-PCR testing was performed with a nasopharyngeal (NP) swab in Heidelberg and India and a combined NP and oropharyngeal (OP) swab in Berlin, as per institutional procedure. Sampling for the Ag-RDT SureStatus was conducted using an NP swab; however, if NP swabbing was contraindicated for clinical reasons (e.g., risk of bleeding), an OP swab was performed. Laboratory personnel working on both the Ag-RDT testing team and the RT-PCR laboratory were blinded to the results of the other test.

**(iii) RT-PCR testing.** The samples collected for routine RT-PCR testing were stored in the provided Amies solution and processed in the referral laboratories following the established laboratory protocols. The RT-PCR assays used as reference standards were the Thermo Fisher TaqPath COVID-19 combo kit PCR on the Applied Biosystems platform (Waltham, MA, USA) in Chennai, India, the Allplex SARS-CoV-2 assay from Seegene (Seoul, South Korea) in Heidelberg, and the Roche Cobas SARS CoV-2 assay (Pleasanton, CA, USA) on the Cobas 6800 or 8800 system or the SARS CoV-2 assay from TIB Molbiol (Berlin, Germany) in Berlin. The Cobas assays apply multiplex RT-PCR with two genomic targets (the ORF1a and E genes), and the TIB Molbiol assay targets the SARS-CoV-2 E gene region. Both test systems comprise internal processing controls to test for inhibition during RNA purification and RT-PCR, as well as positive and negative controls to monitor the overall test performance. A sample was considered positive for SARS-CoV-2 RNA if a specific signal for one, or both in the case of the Cobas systems, was measured and all internal and external controls were valid. More specification (specificity, target specific technical limits of detection [LoD], etc.) of these CE-labeled commercial tests are given in the manufacturer's instructions.

**(iv) Test evaluated.** The test evaluated was the SureStatus COVID-19 antigen card test (Premier Medical Corporation, Mumbai, India). SureStatus relies on the principle of lateral-flow rapid chromatographic immunoassay for detection of the SARS-CoV-2 nucleocapsid protein antigen in a cassette-based format. The test kit includes all required reagents and proprietary swabs for NP/OP sample collection, and the manufacturer's instructions for use (IFU) were followed during the sampling and testing procedures. As indicated in the IFU, 12 drops of extraction buffer are added to a proprietary tube, and the swab is inserted and swirled 5 to 10 times. While squeezing the sides, the swab is removed, and three drops of the specimen are applied to the test device through a nozzle cap. Colloidal gold conjugated antinucleocapsid antibodies on the membrane strip bind to viral antigens, forming an antibody-antigen complex and generating a color change on the test strip that can be interpreted visually after 15 to 20 min.

**(v) Ag-RDT testing.** Ag-RDT testing was performed in immediate proximity to the sample collection for routine RT-PCR testing. The laboratory workstations were kept clean, contaminated materials were kept separate, and the laboratory personnel were trained in handling infectious material. The temperature and humidity at the testing sites were recorded daily. The Ag-RDT test was conducted immediately after sample collection by trained laboratory personnel and interpreted by eye after 15 min. Two readers interpreted each test, both blinded to the results of the other. In case of discrepant results, both readers reinterpreted the results and agreed on one final result. Invalid test results were repeated once with the remaining buffer solution.

**(vi) Clinical data collection.** All participants were asked to provide clinical information about their symptoms, symptom duration, and severity of disease (questionnaire available in Table S2). In Heidelberg, the participants indicated their contact preferences during enrollment and after leaving the drive-in testing site, were contacted by phone or email to complete the questionnaire. In Berlin and India, the questionnaire was completed on-site by a study team member prior to sample collection.

**Data management.** All data were collected and managed using Research Electronic Data Capture (REDCap) tools hosted at Heidelberg University for the Heidelberg and Berlin study sites (22) and the OpenClinica 4 system (Waltham, MA) for the study site in Chennai, India.

**System usability scale and ease-of-use assessment.** A standardized system usability scale (SUS) questionnaire and an ease-of-use (EoU) assessment were designed for this study to evaluate the usability and feasibility of the test and can be found in Questionnaires S3 and S4 in the supplemental material (23). Laboratory personnel from all study sites were invited to complete both questionnaires. An overall SUS score of 68 was interpreted as average (23). A heat map was generated to analyze aspects related to the EoU assessment, categorizing each as satisfactory, average, or dissatisfactory (Fig. S5).

**Statistical analysis.** For the primary analysis, the pooled sensitivity and specificity of the Ag-RDTs were calculated using a fixed-effects model by comparing the Ag-RDT results to RT-PCR as the reference standard. The 95% confidence intervals (CIs) were calculated using Wilson's method. Subgroup analyses combined data from all sites and included symptom duration ($\geq$7 or $<$7 days, $\geq$3 or $<$3 days). Invalid Ag-RDT results were reported separately. Interoperator variability was assessed using Cohen's $\kappa$ statistic to calculate the agreement of positive and negative results between the two independent readers. The analysis was conducted using the statistical software R version 4.03 (R Foundation for Statistical Computing, Vienna, Austria).

For the usability assessment, the SUS score was calculated using the mean value of all answers for each test. For the EoU assessment, responses were scored on a predefined numerical scale, and the mean values were summarized in a heat map. Both assessments were analyzed using Microsoft Excel.

**Data availability.** The deidentified raw data can be requested by contacting the corresponding author.

## SUPPLEMENTAL MATERIAL

Supplemental material is available online only.

**SUPPLEMENTAL FILE 1**, PDF file, 1 MB.

## ACKNOWLEDGMENTS

This study was substantially supported by Angelika Sandritter and the KTS company. Further funding support came from Ministry of Science, Research and Arts, State of Baden-Wuerttemberg, Germany, internal funds from the Heidelberg University Hospital and Charité University Hospital Berlin, the UK Department of International Development, WHO, and Unitaid. The testing devices and all components were procured by FIND. FIND also provided input on study design and data analysis in the form of an academic exchange with the rest of the study group. T.C.J. is in part funded through NIAID-NIH CEIRS contract HHSN272201400008C.

Charité University Hospital Berlin is a corporate member of Freie Universität Berlin and Humboldt-Universität zu Berlin.

The manufacturer and study funders had no input into the study protocol, analysis, or interpretation of results. All authors had access to all data at all times. We declare no competing interests.

J.A.S., C.M.D., N.R.P., and L.J.K. designed the study. L.J.K., S.S., and C.M.D. wrote the manuscript, L.J.K., J.K., and C.M.D. supervised the study site in Heidelberg, O.N. managed all data, F.T. conducted the analysis, F.L. managed the SUS questionnaire and EoU assessment, P.S. conducted RT-PCR testing in Heidelberg, A.K.L. and F.P.M. supervised the study site in Berlin, O.N. was responsible for the Ag-RDT testing, V.M.C. and T.C.J. conducted RT-PCR testing in Berlin, B.K. and A.W. provided all resources in Heidelberg, J.A.S. and M.D.V. supported the study design and manuscript writing, S.O. and S.W. wrote the R code and conducted the meta-analysis, N.S., J.S., H.S., A.P., and M.T. supervised the Indian study site and conducted the RT-PCR testing, and P.L. supported the Indian team. C.M.D. made the final decision to submit the manuscript for publication.

The study team comprised the following members: Kholoud Assaad, Andrea Fuhs, Christopher Harter, Cristopher Schulze, and Gunter Schmitt (Department of Public Health Rhein Neckar Region, Heidelberg, Germany); Martina Fink, Maximilian Schirmer, Annika Small, Matthias Meinlschmidt, Valerie Dürr, Alina Schuckert, Salome Steinke, Henrik Ellinghaus, Alexander Penning, and Loai Abutaima (Division of Infectious Disease and Tropical Medicine, Heidelberg University Hospital, Heidelberg, Germany) Mandy Kollatzsch, Mia Wintel, Franka Kausch, Franziska Hommes, Alisa Bölke, Julian Bernhard, Claudia Hülso, and Elisabeth Linzbach (Institute of Tropical Medicine and International Health, Charité—Universitätsmedizin Berlin, Berlin, Germany); Heike Rössig (Medical Directorate, Charité—Universitätsmedizin Berlin, Berlin, Germany); Maximilian Gertler (Institute of Tropical Medicine and International Health, Charité—Universitätsmedizin Berlin, Berlin, Germany); Susen Burock (Charité Comprehensive Cancer Center, Charité—Universitätsmedizin Berlin, Berlin, Germany); Katja von dem Busche (Department of Pediatric Surgery, Charité—Universitätsmedizin Berlin, Berlin, Germany); and Stephanie Patberg (Berlin Institute for Clinical Teratology and Drug Risk

Assessment in Pregnancy, Institute of Clinical Pharmacology and Toxicology, Charité—Universitätsmedizin Berlin, Berlin, Germany).

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
