## [Reviewer comments · Microbiology Spectrum]

Microbiology Spectrum

A multi-center clinical diagnostic accuracy study of SureStatus - an affordable, WHO emergency-use-listed, rapid, point-of-care, antigen-detecting diagnostic test for SARS-CoV-2

Lisa Krüger, Andreas Lindner, Mary Gaeddert, Frank Tobian, Julian Klein, Salome Steinke, Federica Lainati, Paul Schnitzler, Olga Nikolai, Frank Mockenhaupt, Joachim Seybold, Victor Corman, Terence Jones, Nira Pollock, Britta Knorr, Andreas Welker, Stephan Weber, Nandini Sethurarnan, Jayanthi Swaminathan, Hilda Solomon, Ajay Padmanaban, Ma Thirunarayan, Prabakaran L, Margaretha de Vos, Stefano Ongarello, Jilian Sacks, Camille Escadafal, and Claudia Denkinger

Corresponding Author(s): Claudia Denkinger, Heidelberg University Hospital, Center of Infectious Diseases

Review Timeline:

Submission Date:	April 4, 2022
Editorial Decision:	May 10, 2022
Revision Received:	July 28, 2022
Accepted:	August 4, 2022

Editor: Heba Mostafa

Reviewer(s): The reviewers have opted to remain anonymous.

Transaction Report:

DOI: <https://doi.org/10.1128/spectrum.01229-22>

May 10, 2022

Dr. Claudia M Denkinge
Heidelberg University Hospital, Center of Infectious Diseases
Division of Tropical Medicine
Im Neuenheimer Feld 324
Heidelberg 69120
Germany

Re: Spectrum01229-22 (A multi-center clinical diagnostic accuracy study of SureStatus - an affordable, WHO emergency-use-listed, rapid, point-of-care, antigen-detecting diagnostic test for SARS-CoV-2)

Dear Dr. Claudia M Denkinge:

Link Not Available

Sincerely,

Heba Mostafa

Journals Department
Reviewer comments:

Reviewer #1 (Comments for the Author):

This manuscript reports on large-scale assessment of the performance and ease-of-use of the SureStatus SARS-CoV-2 rapid antigen test in differently resourced settings, including in the global South. It is a strong study with notable attention to detail in the description of study methods and discussion of the results and will make a strong contribution to the literature in the pages of Microbiology Spectrum. The only major points are (1) to clarify the role of this study relative to prior studies regarding SureStatus; (2) to clarify whether this test is meant for self-administration, from what source (NP/AN/OP), and whether this was done; and (3) to provide a little more information about the reference testing systems (limits of detection [LoDs] and ideally also

Ct values at those LoDs). These constitute minor revisions.

Major points:

I54-55: since SureStatus' sensitivity and specificity have already been measured (I51), the authors should here clarify what the contribution of this manuscript is (I believe a much larger, real-world trial). This is not to say the manuscript isn't a valuable contribution; it just needs to be made clear how it is differentiated from what has come before regarding SureStatus (size? multisite? India? independent of manufacturer? etc.). Relatedly, in I86-87: was the antigen-test not self-administered? These lines read as though the laboratory staff collected the test, but as the authors mention one selling point of these tests is self-collection. Please clarify

I understood from the manuscript that nasopharyngeal sampling was for antigen test: but won't it be used in nasal self-swabbing samples? Authors should note this (including if I am wrong; either way it needs clarifying).

I93-101: the limits of detection of these systems, ideally in copies of viral mRNA/mL transfer medium, should be listed in parentheses next to each test. I179-181: How do the Ct value scales compare on the various instruments used? Are they comparable? This is important to mention. I192: This is interesting, and is discussed later (in the Discussion section). While the investigators do mention that a calibration against standards was not possible, I imagine some insight could be gained simply from listing the LoDs of the reference platforms, and the Ct values at the LoDs. It would help the reader decide whether the lower sensitivity in India could be explained by differences in Ct scale (i.e., does a given Ct value correspond to a lower viral load on the testing platforms used in India than in Germany)? The authors should cite Clinical Infectious Diseases 73 (9), e3042-e3046 in this context. The only other major difference I noted in the results was the male-to-female ratio. The discussion in I245-249 does a good job of thoughtfully exploring other potential explanations.

Minor points:

I19: add "when Ct values were low and viral loads were high"

I31, I201, elsewhere: Ct-Values -> Ct values

I76: enrolment -> enrollment?

I80: previous positive RT-PCR test at any time? If so, please mention

I81: list in parentheses acceptable local languages (I assume just German for the German sites, but multiple languages in Chennai, including English? goes to what if any minority populations might have been undercounted)

I94: testing, -> testing

I94: solution, -> solution

I106: please list whether the sample-collection swabs are OP, AN, and/or NP

I119: personnel, -> personnel and

I124: notwithstanding the questionnaire, it would be nice to list at least a few examples of comorbidities in parentheses here

I153: FIND, -> FIND

I149: it might be nice to know whether the results were skewed in any way; reporting at least a median would be helpful (but not required)

I199: decreased -> increased (74.3 to 78.6%)?

I228-230: this is disturbing... Sounds like a bad lot? Were the bent swabs seen at both sites?

I238: a high -> high (multiple high-resource settings were investigated)

I257: standards, -> standards

I265: delta, the -> delta,

I266: suggest -> suggests

I290-291: I see how this is a limitation, but it is also a strength: one can be fairly sure that the positives were not breakthroughs

Staff Comments:

Preparing Revision Guidelines

Please return the manuscript within 60 days; if you cannot complete the modification within this time period, please contact me. If you do not wish to modify the manuscript and prefer to submit it to another journal, please notify me of your decision immediately so that the manuscript may be formally withdrawn from consideration by Microbiology Spectrum.

Reviewer comments:

Reviewer #1	
To clarify whether this test is meant for self-administration, from what source (NP/AN/OP), and whether this was done	Thank you for this comment. This test has been studied using NP/OP sample collection as described in the section study procedures (Lines 84-93).
To provide a little more information about the reference testing systems (limits of detection [LoDs] and ideally also Ct values at those LoDs.	Thank you for this comment. We have revised and rewritten the section RT-OCR Testing (96-111) to give more information for the standards used.
I54-55: since SureStatus' sensitivity and specificity have already been measured (I51), the authors should here clarify what the contribution of this manuscript is (I believe a much larger, real-world trial). This is not to say the manuscript isn't a valuable contribution; it just needs to be made clear how it is differentiated from what has come before regarding SureStatus (size? multisite? India? independent of manufacturer? etc.).	As suggested by the reviewer we have included more detail about the value of this study in the introduction in adding the following sentence: "This study represents the first multi-center, manufacturer independent diagnostic accuracy study for the Ag-RDT SureStatus." (Lines 57-58).
a) Relatedly, in I86-87: was the antigen-test not self-administered? These lines read as though the laboratory staff collected the test, but as the authors mention one selling point of these tests is self-collection. Please clarify b) I understood from the manuscript that nasopharyngeal sampling was for antigen test: but won't it be used in nasal self-swabbing samples? Authors should note this (including if I am wrong; either way it needs clarifying).	Thank you for this comment. This study collected the sample NP or if contraindicated OP as explained in the section study procedures (Lines 84-93). In the discussion (Lines 273-277) we refer that AN sampling has been studied as a good alternative to NP/OP sampling and could be used if a study confirms the high sensitivity with AN sampling. We have rephrased a sentence in the discussion to make the sampling method used clearer: "Furthermore, given that self-sampling from the anterior nose (AN) is a reliable alternative to professional nasopharyngeal sampling, scale-up of testing could be possible without requiring large numbers of trained health-care workers, provided the result obtained with NP/OP sample collection here can be confirmed with AN sampling." (Line 276).
I93-101: the limits of detection of these systems, ideally in copies of viral mRNA/mL transfer medium, should be listed in parentheses next to each test.	In combination with another comment, we have revised and rewritten the section RT-OCR Testing (96-111) to give more information for the standards used.
I179-181: How do the Ct value scales compare on the various instruments used? Are they comparable? This is important to mention.	Thank you for this comment. As mentioned in Line 287-290 in the section Discussion this is one limitation of our study. Using different PCR references creates a limitation in the comparison of the Ct-values for analysis, however due to the pandemic situation and shortages of supplies, we

	were not able to only have one PCR reference test.
l192: This is interesting, and is discussed later (in the Discussion section). While the investigators do mention that a calibration against standards was not possible, I imagine some insight could be gained simply from listing the LoDs of the reference platforms, and the Ct values at the LoDs. It would help the reader decide whether the lower sensitivity in India could be explained by differences in Ct scale (i.e., does a given Ct value correspond to a lower viral load on the testing platforms used in India than in Germany)?	In combination with another comment, we have revised and rewritten the section RT-OCR Testing (96-111) to give more information for the standards used.
l19: add "when Ct values were low and viral loads were high"	Thank you, we have completed the sentence with your suggested addition.
l31, l201, elsewhere: Ct-Values -> Ct values	We have changed Ct-Values (capital V) in the whole manuscript to Ct-values (lower v).
l76: enrolment -> enrollment?	Thank you for this comment. In our understanding enrolment is the right grammatical usage and therefore we have not changed this wording as suggested by the reviewer.
l80: previous positive RT-PCR test at any time? If so, please mention	Thank you for this comment, we have included the detail that the positive RT-PCR test did not have any time limit and could have been at any time during the pandemic.
l81: list in parentheses acceptable local languages (I assume just German for the German sites, but multiple languages in Chennai, including English? goes to what if any minority populations might have been undercounted)	Thank you for this comment. We have added the languages as recommended by the reviewer in parentheses (Lines 82-83).
l94: testing, -> testing	We have corrected this mistake, thank you for this recommendation.
l94: solution, -> solution	Thank you for recognizing this spelling mistake – corrected as suggested by the reviewer.
l106: please list whether the sample-collection swabs are OP, AN, and/or NP	We have included more information on the sample collection in this section and would like to point out that in the section study procedures a detailed description of the sampling for Ag-testing and RT-PCR-testing (85-93) is included.
l119: personnel, -> personnel and	Thank you for this suggestion, we have included it as recommended.
l124: notwithstanding the questionnaire, it would be nice to list at least a few examples of comorbidities in parentheses here	Thank you for this suggestion. We have decided to delete the comorbidity part from the questionnaire (see supplement with already updated version) as not all study sites had the resources to document this information throughout the whole study period. We have missed to update the manuscript in this section

	and thank you for your comment.
l153: FIND, -> FIND	Thank you, we have updated this spelling mistake.
l149: it might be nice to know whether the results were skewed in any way; reporting at least a median would be helpful (but not required)	Thank you for this comment, unfortunately we will not be able to report the median and believe this will not influence the submission.
l199: decreased -> increased (74.3 to 78.6%)?	Thank you for the comment. We have changed the sentence to make it more readable and understandable. The comparison was made to the sensitivity of Germany and participants with a Ct-value ≤ 30 .
l228-230: this is disturbing... Sounds like a bad lot? Were the bent swabs seen at both sites?	This quality problem was only observed in Germany and reported back to the manufacturer. In India this problem was not experienced.
l238: a high -> high (multiple high-resource settings were investigated)	Thank you for this suggestion, the manuscript has been changed accordingly.
l257: standards, -> standards	The spelling mistake has been changed – thank you.
l265: delta, the -> delta,	As recommended by reviewer the comma was deleted in the manuscript.
l266: suggest -> suggests	The manuscript has been changed according the suggestion by the reviewer.
l290-291: I see how this is a limitation, but it is also a strength: one can be fairly sure that the positives were not breakthroughs	Thank you for this comment. We agree that this could be seen as both advantages and disadvantage and will leave this section as an important focus for further research in our manuscript.

August 4, 2022

Dr. Claudia M Denkinger
Heidelberg University Hospital, Center of Infectious Diseases
Division of Tropical Medicine
Im Neuenheimer Feld 324
Heidelberg 69120
Germany

Re: Spectrum01229-22R1 (A multi-center clinical diagnostic accuracy study of SureStatus - an affordable, WHO emergency-use-listed, rapid, point-of-care, antigen-detecting diagnostic test for SARS-CoV-2)

Dear Dr. Claudia M Denkinger:

Your manuscript has been accepted, and I am forwarding it to the ASM Journals Department for publication. You will be notified when your proofs are ready to be viewed.

Sincerely,

Heba Mostafa
Editor, Microbiology Spectrum
